

# About right: references in Open Access EGU journals

Andrea Pozzer[1]

[1]Max Planck Institute for Chemistry, 55128, Mainz, Germany

**Correspondence:** Andrea Pozzer (andrea.pozzer@mpic.de)

**Abstract.** We investigated the number of references per page for different European Geophysical Union journals, which share the same text formatting. Although the journals formally all focus on geoscience, different disciplines are accommodated, from ocean science and biogeosciences to the technical description of numerical model development. Here, we show that across such different disciplines, the number of references per page is remarkably constant, revealing that there is a consensus regarding optimum reference density in the community. Further, this value has remained constant in the last decade, despite the consistent increase in the number of pages and in the number of references in almost all journals investigated. Independently of the quality of the references used in a article, we show that for the EGU journals the average number of references per page is 3.82 ($1.87 - 6.18$ at 90% confidence level).

## 1 Introduction

The number of references in a scientific paper can influence reader judgement of the paper's quality (Lovaglia, 1991), and is thus an important factor in defining its success, i.e. its number of citations (Fox et al., 2016). Therefore it is important that authors include an optimal (and balanced) quantity and quality of references in their articles.

It has been shown (Abt and Garfield, 2002) that the number of references per unit page is remarkably constant across a large number of disciplines, and that longer papers are, on average, more cited than shorter papers (Leimu and Koricheva, 2005). Nevertheless, the creation of homogeneous and standardised text length is a challenging task, with each journal having different formatting layouts, which could influence the perception of reference quantity and, indirectly, result in pressure for an increase or decrease in their numbers.

The European Geophysical Society (one of the predecessors of the European Geophysical Union), started its first open access (OA) journal in 2001, with the launch of the *Atmospheric-Chemistry and Physics* journal (Pöschl, 2004, 2012). The success of this first journal prompted The European Geophysical Union (EGU), through Copernicus Publications, to establish additional OA journals; 19 journals are currently published by Copernicus (for EGU), covering various topics of the Earth, planetary and space sciences.

In this work we took advantage of the OA EGU journals, which have identical layouts, and therefore allow for a direct comparison between the different journals. In addition, all the metadata of the papers have been published online in a searchable xml format, which allows automatic computer scripting for information gathering. It must be stressed that Copernicus





Publications publish other OA journal in addition to the EGU journals studied. However, these journals use diverse layout, and the comparison between such different formats is not always straightforward.

In this work we analyse the reference density, i.e. the number of references per page, in the OA journals published by the EGU. We show that there exists a sound range for the number of references per page, and that this has remain remarkably constant over time. In the Sec. 2, the methods for data collection is explained, followed by an analysis of the temporal trends (Sect. 3) is presented. Finally, the main results are derived in Sec. 4, followed by the conclusions.

## 2 Methodology

We considered articles accepted and published in xml form in the 2010–2020 period from the EGU OA journals. Therefore, only EGU journals which started operating in 2010 at latest were used in this study, which resulted in the inclusion of a total of 12 journals (see Table 1):

- *GMD, Geoscientific Model Development*,

- *ACP, Atmospheric Chemistry and Physics*,

- *BG, Biogeosciences*,

- *CP, Climate of the Past*,

- *AMT, Atmospheric Measurement Techniques*,

- *OS, Ocean Science*,

- *ESD, Earth System Dynamics*,

- *TC, The Cryosphere*,

- *NHESS, Natural Hazards and Earth System Sciences*,

- *NPG, Nonlinear Processes in Geophysics*,

- *SE, Solid Earth*,

- *HESS, Hydrology and Earth System Sciences*.

An automatic python script was used to recursively collect all the needed information, such as the number of pages and the number of references, from the XML version of each manuscript.

To avoid counting papers which cited an unrepresentative number of references (such as zero references or pure compilation articles), the outliers, which were defined as (i) papers containing no references or (ii) the papers containing a number of references above the average plus 3 times the reference's standard deviation, were removed. In total 30,028 papers were downloaded, of which 787 were excluded as outliers, i.e. 29,241 published papers were used in this analysis.





In Table 1 the total number of papers analysed and those excluded from the analysis for each journal are presented. Roughly,
$\simeq$ 3 % of the papers published in each journal were excluded as outliers. The outlier fraction is almost constant among all journals, ranging from 1.7% for TC to 3.6% for NHESS.

**Table 1.** Summary of journal characteristics.The number of papers analysed in each journal are listed, as well the number excluded (also expressed as a fraction) as outliers. The papaers with the highest number of pages and highest numbers of references are also listed for each journal.

| Journal | papers | excluded | % excluded | highest num.pages | | highest num. refs | |
|---|---|---|---|---|---|---|---|
| GMD | 1931 | 45 | 2.3 | 66 | gmd-13-3643-2020 | 526 | gmd-12-3149-2019 |
| ACP | 8965 | 184 | 2.1 | 583 | acp-15-4399-2015 | 793 | acp-15-4399-2015 |
| BG | 4197 | 147 | 3.5 | 56 | bg-11-3547-2014 | 469 | bg-7-2851-2010 |
| CP | 1318 | 35 | 2.7 | 50 | cp-13-1851-2017 | 531 | cp-13-1851-2017 |
| AMT | 3210 | 80 | 2.5 | 75 | amt-9-4181-2016 | 346 | amt-12-525-2019 |
| OS | 849 | 26 | 3.1 | 51 | os-6-91-2010 | 255 | os-14-471-2018 |
| ESD | 468 | 11 | 2.4 | 74 | esd-10-379-2019 | 572 | esd-10-379-2019 |
| TC | 1757 | 30 | 1.7 | 47 | tc-13-3261-2019 | 402 | tc-12-759-2018 |
| NHESS | 2500 | 91 | 3.6 | 42 | nhess-18-2561-2018 | 308 | nhess-11-2617-2011 |
| NPG | 664 | 17 | 2.6 | 56 | npg-18-295-2011 | 418 | npg-18-295-2011 |
| SE | 791 | 26 | 3.3 | 49 | se-7-1417-2016 | 293 | se-5-1243-2014 |
| HESS | 3378 | 95 | 2.8 | 47 | hess-23-303-2019 | 431 | hess-20-3799-2016 |

In addition to the number of analysed and disregarded papers, Table 1 lists the papers with the highest numbers of pages and references in the period 2010–2020 for each journal. The longest articles range in length from 42 pages (*nhess-18-2561-2018*) to 583 pages (*acp-15-4399-2015*). The maximum number of references in an article ranges from 255 *os-14-471-2018* to 793
in *acp-15-4399-2015*. *acp-15-4399-2015* stands out among all other EGU articles both with respect to its number of pages as well as the number of references. In this paper, a list of measurements of Henry's law coefficient for numerous gases of atmospheric relevance are presented. However, it should be noted that not all the papers with higher numbers of references are review articles or compilations of measurements (see for example, *amt-12-525-2019* or *gmd-12-3149-2019*).

## 3  Temporal trends

In the last decades, the length of scientific papers undergone a significant increase. (Ucar et al., 2014) showed not only a clear trend towards an increase in the number of pages in papers in engineering journals, but also showed that this increase has not yet begun to level off.

This increase in paper length is mirrored by a constant increase in the number of references over time (Biglu, 2008; Jaunich, 2018). Bornmann and Mutz (2015) revealed a large increase in the number of references from the middle of the twentieth
century onward. The temporal increase in the number of references per papers varies among different disciplines (Sánchez-





Gil et al., 2018). Furthermore, Nicolaisen and Frandsen (2021) showed that "there is a drop in short reference lists and a corresponding increase in a bit longer and medium size reference lists. Long and very long reference lists remain much more stable in shares over time, and does therefore not contribute much to the observed growth." A steady state in reference numbers has until now only been artificially reached in a few journals and/or manuscript types, through the enforcement of limits in

the number of references (Anger, 1999). Nevertheless, most of these studies only focused on the number of references per article, without analysing this parameter with respect to the paper length, or, similarly, without investigating reference density. A notable exception is the work of Milojević (2012), which found different temporal trends in reference per page, depending on the field of study.

We estimated for each analysed journal the trends in number of pages, references and references per page, and our results are

presented in Table 2. In EGU publications, the number of pages and references per paper have been increasing in the last decade. The increase in pages per paper ranges from 0.26 $\mathrm{pages/yr}$ in ESD to 0.90 $\mathrm{pages/yr}$ in SE. Similarly, the number of references also increased in the same period, ranging from 1.06 to 3.91 $\mathrm{references/yr}$ in ESD and SE, respectively. Importantly, all these estimated temporal trends (both number of pages and number of references) are statistically significant at 99% confidence level, with the exception of the ESD journal.

**Table 2.** Linear fit of the temporal trends of pages, references (column refs) and references per page (column refs/page) for different EGU journals for all analysed papers between 2010 and 2020. The numbers inside the parentheses are the standard deviations of the estimated time trends (slope of the linear fit). The units are in $\mathrm{yr}^{-1}$.

| Journal | pages | refs | refs/pages |
|---|---|---|---|
| GMD | 0.52 ( 0.06) | 2.01 ( 0.27) | 0.02 ( 0.01) |
| ACP | 0.34 ( 0.02) | 1.81 ( 0.09) | 0.03 ( 0.01) |
| BG | 0.38 ( 0.02) | 1.86 ( 0.14) | 0.01 ( 0.01) |
| CP | 0.43 ( 0.04) | 2.43 ( 0.30) | 0.03 ( 0.01) |
| AMT | 0.39 ( 0.03) | 1.27 ( 0.12) | 0.01 ( 0.01) |
| OS | 0.31 ( 0.05) | 1.61 ( 0.22) | 0.03 ( 0.01) |
| ESD | 0.26 ( 0.10) | 1.06 ( 0.52) | 0.01 ( 0.03) |
| TC | 0.45 ( 0.04) | 1.99 ( 0.18) | 0.02 ( 0.01) |
| NHESS | 0.56 ( 0.03) | 2.34 ( 0.13) | 0.03 ( 0.01) |
| NPG | 0.50 ( 0.06) | 1.04 ( 0.26) | -0.04 ( 0.02) |
| SE | 0.90 ( 0.08) | 3.91 ( 0.39) | -0.00 ( 0.02) |
| HESS | 0.51 ( 0.02) | 2.28 ( 0.13) | 0.03 ( 0.01) |
| total | 0.45 ( 0.01) | 1.90 ( 0.05) | 0.01 ( 0.01) |

The increase in number of citations may be attributed to the increasing growing of available literature. In fact, by publishing more papers, more manuscript can (or must) be cited in future work. Analogously, the increase in absolute number of citation reflects also the maturity that a specific science field has reached, where the large (and increasing) number of citations mirrors





the large (and increasing) research performed on specific topic. Further, accessibility could be a major point for increasing citations over time: OA papers (with the leading role of pure OA journals) allow an easy access to citable material, which can
be easily referred to. In addition, technological development (e.g. fast internet connection, searchable and online downloadable journals) allows favourably the search and usage of precedent literature. Finally, Persson et al. (2004) suggested that with the intensification of scientific collaboration an increase of citations of co-published paper must be expected, and therefore this increase is a sign of increasing national and international collaboration between research teams.

In addition to the increase in the number of pages and the number of references in the period 2010-2020, we estimated also
the evolution of reference density over this period. As shown in Table 2, these trends are very close to zero. The only journal with a clear statistically trend is ACP, which present an increase of references per page per year equal to 0.032,while none of the other journals present a statistically significant trend. This is in contrast to the findings of Ucar et al. (2014), which found a variable references to pages ratio along the 50 years of study, but is in agreement with the work of Abt and Garfield (2002).

Based on these estimates, we can consider reference density to be constant in OA EGU journals,which will allow us to use
all papers published in the considered period to estimate it.

## 4   Results

The probability density distribution of pages against references is presented in fig. 1. Both pages and references exhibit a clear log-normal distribution, although for a few journals (e.g ESD) the number of papers available was quite low, which complicated the estimation of a meaningful statistics. In each plot the linear fit (with no intercept) was also overlaid on the distribution. The
linear fits range from 2.8 (AMT) to 4.6 (CP) references/page, showing quite homogeneous behaviour within all the papers, with a coherent and similar reference density in all EGU journals.

For each journal the average number of pages and the number of references have been calculated , and the results are presented in Table 3. The average number of pages and references can exhibit strong variations between the journals, with differences of up to 60%. The longest papers appear on average in GMD, with 19 pages, while the shortest were published in
NPG with 12 pages. NPG also exhibit the lowest number of references per paper (i.e. 40 references per paper), while CP has the highest, with 77 references per manuscript on average.

Finally, the average reference densities for each journals (based on the reference density for each manuscript) have been estimated (see Table 3 and fig. 2). The number of references per page ranges from 3.00 to 4.77, for AMT and CP, respectively. Despite the differences in reference number or page distribution between the journals, the numbers of reference per page are
statistically similar for all journals.

We have seen that the reference density for each journal presents a classical log-normal distribution. We then combined all the reference density distributions, obtaining, in the first order approximation, also a log-normal distribution (Mitchell, 1968; Cobb et al., 2012; Dufresne, 2008). We therefore estimated the overall reference density obtaining an average of 3.82 references/page with a confidence level of 90% between 1.87 and 6.18 references/page.





**Figure 1.** 2-Dimensional histogram (center) with frequency histogram for pages (top) and references (right) for different EGU journals. The journal name and the total number of papers, pages and references are listed on the top right of each plot. The black line depict the linear fit (with no intercept). The axes for the 2-dimensional histograms are the same in all plots.





**Table 3.** Average numbers of pages, references (column refs) and references per page (column refs/page) for different EGU journals for all analysed papers. The range at 90% confidence level is listed in parentheses.

| Journal | papers | pages | refs | refs/pages |
|---|---|---|---|---|
| GMD | 1886 | 19 ( 10- 32) | 65 ( 22- 129) | 3.37 ( 1.55- 5.50) |
| ACP | 8781 | 17 ( 10- 27) | 65 ( 30- 114) | 3.90 ( 2.11- 6.00) |
| BG | 4050 | 15 ( 9- 24) | 70 ( 35- 118) | 4.57 ( 2.61- 6.75) |
| CP | 1283 | 16 ( 9- 25) | 76 ( 32- 141) | 4.77 ( 2.50- 7.38) |
| AMT | 3130 | 15 ( 9- 26) | 46 ( 19- 84) | 3.00 ( 1.45- 4.94) |
| OS | 823 | 15 ( 8- 24) | 49 ( 21- 90) | 3.34 ( 1.67- 5.50) |
| ESD | 457 | 16 ( 9- 25) | 61 ( 25- 117) | 3.88 ( 1.94- 6.24) |
| TC | 1727 | 15 ( 8- 24) | 56 ( 23- 97) | 3.68 ( 2.07- 5.56) |
| NHESS | 2409 | 13 ( 6- 22) | 45 ( 16- 86) | 3.48 ( 1.62- 5.80) |
| NPG | 647 | 12 ( 6- 21) | 40 ( 15- 74) | 3.43 ( 1.59- 5.70) |
| SE | 765 | 16 ( 8- 28) | 68 ( 27- 129) | 4.39 ( 2.14- 7.07) |
| HESS | 3283 | 16 ( 9- 24) | 58 ( 26- 103) | 3.70 ( 1.94- 5.86) |
| total | 29241 | 16 ( 9- 26) | 60 ( 24- 111) | 3.82 ( 1.87- 6.18) |

It is difficult to establish the cause of the relationship between pages and references, although it is clear that number of pages and number of references in a paper not only influence positively each other but they are influenced directly and indirectly by multiple factors Abt and Garfield (e.g. number of authors 2002). Nevertheless, here we showed that the journal layout play an essential role in defining this ratio, as this remains constant between all the OA EGU journals, independently on the research field, therefore substantially confirming the findings of Abt and Garfield (2002).

## 5   Conclusions

The importance of references in scientific journals has been clearly established. In this work we took advantage of the OA EGU journals, which, although they cover different areas in geoscience, share the same layout, thereby allowing for a direct comparison. It is shown that in the period 2010-2020, the number of pages and the number of references has been increasing in a statistically significant way.

Different reason could be behind this growth, such as open access to existing literature together with technological development which allows easier research of relevant citations. Additionally, we suggested this growth to be especially strong in EGU journals, being geophysics still a quite novel science with strong growing research and, consequently, strong growing of published literature, which tends to be more and more referenced in following studies.

Despite the increases in publication length and number of references in all journals since 2010, the reference density (i.e. 135   number of references per page) has remained remarkably constant. In addition, no statistical difference in reference density can be observed in any of the journals. The average number of references per published page has been estimated based on all



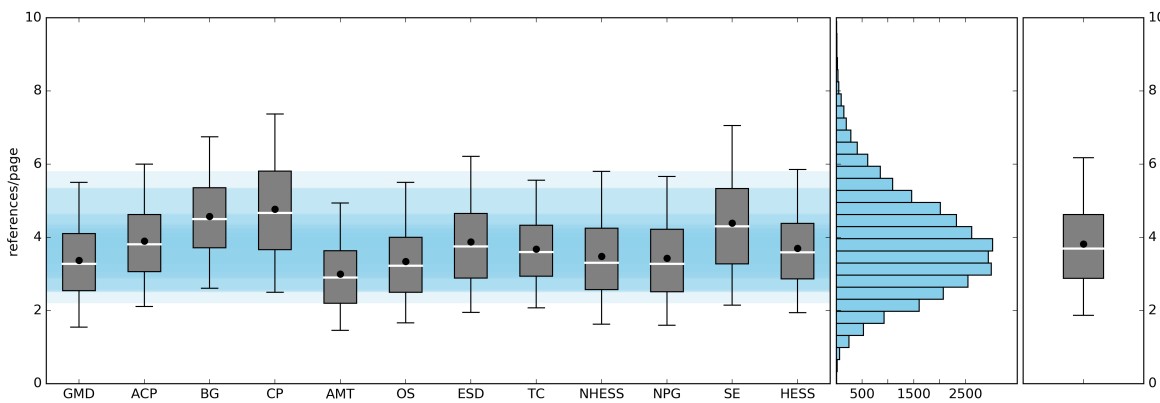

**Figure 2.** Left: Box plot of numbers of references per page. The box represents the distribution quartiles (25% and 75%), the white lines are the medians, and the black dots the averages. The bars represent the 90% confidence levels. The acronym of the respective journal is listed on the bottom. The light blue area represents the overlap of the 25-75% quartiles range for all the journals. Middle: Probability density histograms of numbers of references per page for all the papers from all journals. Right: Box plot of numbers of references per page as on the left, but for all papers from all journals.

the published papers, which show that the optimal reference density is 3.82 references/page ($1.87 - 6.18$ at 90% confidence level). This work shows that the layout does influence the number of references per page, confirming previous work

It has been shown that papers with high number of references tends to be cited more (Lovaglia, 1991); here we showed that these correlate with longer papers, suggesting that papers presenting the work in more detail and with greater availability of data or ideas tend to have greater impact on following literature. Therefore, pages or references limit should be strongly avoided in journals, as authors could be discourage to describe their study with sufficient details. This philosophy is also behind the OA EGU journals, which do not impose any limit to their published papers, which can be as long as they need to be.

*Code availability.* The code for analysis is available upon request to the contact author.

*Author contributions.* A.P. designed the study, performed the calculations and analysed the results.





*Competing interests.* No competing interests are present.

*Acknowledgements.* The author acknowledge the constructive discussion with U.Pöschl



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
