# Peer review of "About right: references in Open Access EGU journals"

_Geoscience Communication, 2021_

## Author Comment (AC1)

I thank the referee #1 for the review.
In the following, the comments are repeated (in bold) with my reply.

1. **I find the manuscript quite long and detailed in reation for the resulting information given. Is it really necessary to go into so much detail? At the end the information gleaned is, while intersting, rather limited. To cut this in half would retain the information, withot the need for extensive statistics and figures.**

   I disagree with the referee on this point, as I believe that these information are necessary, for different reasons. A major principle of science is the "reproducibility". A high number of details would facilitate any replication of this study and its results. Therefore I believe there is no such "too much details", as these could be helpful for some readers. Furthermore, as briefly discussed in this work, the presence of detailed description could spark new ideas, by, for example, modifying the methodology of the work.

   On the other side, I agree with the referee that it is important to convey the message efficiently, especially in scientific papers, and that numerous details could be tedious for the reader. For that reason I consider the abstract the most important part of any papers, where, in few lines, the work is presented and the important results are listed. In summary, the presence of a well written abstract would shorten considerably the needs to read the entire manuscript, focusing only on the "take home message". Only interested readers would then focus on the manuscript itself and it methodological contents, and, for them, no detail is too much. I have revised the abstract so to have a clear description of the results obtained in this work.

2. **The study is limited to only online journals of one publisher; this excludes a large number of other journals from other publishers. Not everybody in the geosciences community publishes in these sorts of journals.**
   Indeed the referee is correct, as I mentioned this in the paper (see line 25). Nevertheless, I also clearly stated that I focused on these journals as they are sharing exactly the same format/layout, therefore allowing a realistic comparison between them (see line 15). This removes any uncertainties or additional errors in converting the different layouts. Furthermore, this gives the opportunity to focus largely on Open Access EGU journal's statistics, showing interesting data on these publications.

3. **The point is made that these online jozrnals do not have length limits – this might be part of the reason or problem**

**in itself; there is a good reason why some of the high-profile journals ("Geology" comes to mind) have length limits. It might be interesting to look at the statistics there. In that context I always have to think of the saysing, often ascribed to B. Pascal, that "I am sorry to have written a long letrer because I did not have enough time to write a short letter". Long is not always better, and being verbose does not neccessarily convey more information, or do it better. The current paper is a case in point.**

I agree with the referee that longer papers do not imply better quality. As the referee correctly mentioned "Long is not always better, and being verbose does not neccessarily convey more information, or do it better." Nevertheless I would like to argue that also short papers do not imply better quality as well, and that "short is not always better". For that reason I ended the paper mentioning that papers should "be as long as they need to be"

I fully agree that most of the high profile journals have a somewhat different approach, but, I would like to argue that the short papers policy in high-profile journals is a consequence, rather than a cause. In fact, due to their popularity and publication requests, high-profile journals can select only the best study for publications, therefore fostering their high-profile and must enforce page limitation to be able to condense the high level publications in the volume/number established length. Furthermore, in high-profile journals the (detailed) methodology is normally presented at the end of the paper in an electronic supplement, therefore using comparable similar page numbers to any other journals (see for example the last volume of "Geology", where almost the totality of the research articles present an electronic supplement, sometimes above 30 pages long). On the other side, online journals do not normally have page restriction, therefore allowing a complete methodology description within the paper itself.

I believe that the two approaches (page limitation and electronic supplement against no page limitation) are conveying fundamentally the same information amount, although with a different structure. It is important to notice that EGU journals allow both approaches, as no page limitation is present and electronic supplements are also permitted.

4. **I am also missing a clear statement what the scientific question or working hypothesis is here? With which scientific goal was this study done? And in the end... what came out of it? Surely the recommendation to not limit the length of a**

**paper can be all of it?** I apology that our scientific hypothesis was not clearly stated enough. Beside the pure statistical analysis (already interesting per se) the main goal of the present paper is to investigate if publications in different sectors, but with the same layout, would have the same citation per page density. This has been indeed confirmed by our study. I will highlight the scientific goals again in the abstract and introduction of the paper.

---

## Author Comment (AC2)

I thank the reviewer for her/his positive comments.

1. **Although the case study is well defined considering only the journals of a specific publisher within a well-determined time frame, a clear description of the research question is lacking. Is this just a statistical analysis or does it provide implications on the publication of papers in these journals (as it seems from the conclusions)?.**

   Indeed, this comment is similar to the one from referee #1, and we agree that our scientific goals were not correctly described. We have tried to highlight them more in the revised version. Beside the pure statistical analysis (already interesting per se) the main goal of the present paper is to investigate if publications in different sectors, but with the same layout, would have the same references per page density. This has been indeed confirmed by our study.

2. **Stating that "pages or references limit should be strongly avoided in journals, as authors could be discouraged to describe their study with sufficient details" is quite a strong conclusion that requires a further discussion and a deeper analysis. Furthermore, given the analysed data, which refer only to some specific EGU Open Access journals in a specific time frame, it is not possible to get to this general conclusion;**

   I agree with the referee this to be a quite strong statement, and indeed this is not fully correct. Based on the reply to referee #1, in fact, many journals with page restrictions do allow (electronic) supplements where additional details (or data) can be listed. Therefore I decided to remove this sentence/conclusion which is not confirmed by any results of this paper.

3. **Although the text is overall well written and easy to read, a better distribution of the information has to be considered. For example, the first paragraphs of the section "Temporal trends" would better fit the "Introduction" section. Moreover, once the research question is clearly defined, more space could be given to the discussion of the results, which here seems quite limited.**

   I thank the referee for pointing this out. I have moved the first paragraph of "Temporal trends" under the "Introduction", and I tried to clarify our objective even more in the paper (see reply to first question). I am however hesitant to discuss more results than the one

presented here, as I believe that more in-depth analysis, with a larger body of journals, would be necessary to draw any general conclusions outside the one already mentioned in the paper.

All the technical corrections have been implemented in the revised version of the manuscript.

---

## Author Response (AR1)

Dear editor,

I have improved the manuscript based on the comments of the reviewers. The main comment, raised by both referee, is that the scientific goals were not clear enough. We therefore restructure the abstract and added a sentence in the introduction, making the main goal of the paper clearer.

In this document, I have listed the changes applied to the manuscript. The manuscript with highlighted changes has been included here as well (from Page 2).

**Abstract** Following recommendation of referee #1 and #2, I have reviewed the abstract. I hope it is now more readable and clearer for all readers. All main conclusions reached in this work are here summarized.

**Introduction** based on the comments of referee #2, I have moved part of the Sect.3 to the Introduction and added a sentence on the goal of this work.

**Conclusions** I reformulated the last sentences of the conclusions, based on the comments of referee #2, removing any suggestion that page limitation could be a problem for the authors, as this is incorrect.

In addition, all minor changes suggested by referee #2 were adopted in the paper.

Best regards,

Andrea Pozzer

[revised manuscript text omitted]

---

## Author Response (AR2)

Dear editor,

I improved the manuscript based on the comments :

**Language** The text was checked by native speaker and it was modified accordingly.

**Conclusions** I added general conclusions for authors and editors/reviewers.
*" This work provides an indication for authors preparing their manuscript for EGU journals, suggesting how many references are "about right" in a paper. This is especially important for less experienced authors, as it shows if their citation strategy fits with the existing body of literature. Furthermore, reviewers or editors should be particularly careful in evaluating manuscript whose reference density is outside the range $1.87 - 6.18$, as this indicate a non-standard (or outlying) manuscript with an uncommonly high (or low) number of references. "*

Best regards,

Andrea Pozzer